

# Targeted metabolomics study of serum bile acid profile in patients with end-stage renal disease undergoing hemodialysis

Rong Li[1,*], Li Zeng[1,2,*], Shuqin Xie[1], Jianwei Chen[1], Yuan Yu[1] and Ling Zhong[1]

[1] Department of Nephrology, Second Affiliated Hospital of Chongqing Medical University, Chongqing, China
[2] Chongqing Key Laboratory of Ultrasound Molecular Imaging, Ultrasound Department of the Second Affiliated Hospital of Chongqing Medical University, Chongqing, China
* These authors contributed equally to this work.

## ABSTRACT

**Background:** Bile acids are important metabolites of intestinal microbiota, which have profound effects on host health. However, whether metabolism of bile acids is involved in the metabolic complications of end-stage renal disease (ESRD), and the effects of bile acids on the prognosis of ESRD remain obscure. Therefore, this study investigated the relationship between altered bile acid profile and the prognosis of ESRD patients.

**Methods:** A targeted metabolomics approach based on ultra performance liquid chromatography-tandem mass spectrometry (UPLC-MS/MS) was used to determine the changes in serum bile acids between ESRD patients ($n = 77$) and healthy controls ($n = 30$). Univariate and multivariate statistical analyses were performed to screen the differential proportions of bile acids between the two groups.

**Results:** Six differentially expressed bile acids were identified as potential biomarkers for differentiating ESRD patients from healthy subjects. The decreased concentrations of chenodeoxycholic acid, deoxycholic acid and cholic acid were significantly associated with dyslipidemia in ESRD patients. Subgroup analyses revealed that the significantly increased concentrations of taurocholic acid, taurochenodeoxycholic acid, taurohyocholic acid and tauro α-muricholic acid were correlated to the poor prognosis of ESRD patients.

**Conclusions:** The serum bile acid profile of ESRD patients differed significantly from that of healthy controls. In addition, the altered serum bile acid profile might contribute to the poor prognosis and metabolic complications of ESRD patients.

Corresponding authors
Yuan Yu,
yuyuan@hospital.cqmu.edu.cn
Ling Zhong,
zhongling@hospital.cqmu.edu.cn

## INTRODUCTION

Chronic kidney disease is an emerging health problem in China, with an overall prevalence of 10.8%. End-stage renal disease (ESRD) accounts for 0.03% of the Chinese population, and the total number of ESRD patients has reached approximately 3.3 million (*Zhang et al., 2012*). Loss of renal function in ESRD patients can result in the abnormal

metabolism of proteins, fatty acids, carbohydrates, vitamins, water, electrolytes and acid-base, leading to dysfunction of various organ systems and poor quality of life. In recent years, the concept of intestinal-renal axis has been increasingly accepted (*Evenepoel, Poesen & Meijers, 2017*), which refers to the interaction between chronic kidney disease and endogenous intestinal microbiota as well as their metabolites that may affect the pathophysiological state of ESRD patients. On one hand, uremia affects the composition and metabolism of intestinal microbiota (*Vaziri et al., 2013*; *Wang et al., 2012*; *Wong et al., 2014*); on the other hand, disordered intestinal microbiota increases the production of enterogenous urotoxin and impairs the barrier function of intestinal mucosa, thereby aggravating renal microinflammation and dyslipidemia, promoting the progression of chronic kidney disease, and elevating the risks of atherosclerosis, cardiovascular and cerebrovascular diseases (*Evenepoel, Poesen & Meijers, 2017*; *Vaziri, Zhao & Pahl, 2016*). Hence, enterogenous metabolites have become an important research topic for combating the metabolic complications of chronic kidney disease.

Bile acids are essential metabolites of intestinal microbiota. Primary bile acids (PBA) are initially synthesized from cholesterol (CHOL) in the liver (*Javitt, 1994*), and then conjugated with taurine or glycine to form conjugated bile acids (ConBA) (*Russell, 2003*), which are secreted into the intestine via the bile duct. These bile acids are mostly reabsorbed at the end of the ileum, and re-transferred through the portal vein back to the liver, thereby forming a complete enterohepatic circulation of bile acids. The remaining small amount of unreabsorbed bile acids may undergo a series of biotransformation under the action of intestinal microbiota, such as deconjugation, dehydroxylation, dehydrogenation and epimerization, to produce secondary and tertiary bile acids (*Midtvedt, 1974*). Bile acids have been known to play a pivotal role in promoting the absorption of lipids and fat-soluble nutrients in intestinal tract. More recently, they have also been found to affect lipid, glucose and energy homeostasis by acting as endogenous signaling molecules conjugated to bile acid receptors, including farnesoid X receptor (FXR) and G protein-coupled bile acid receptor (GPBAR) (*Hylemon et al., 2009*).

At present, ESRD is commonly diagnosed by glomerular filtration rate test ($\leq$15 mL/min/1.73 m$^2$). Considering the recent advancements in various "omics" fields such as genomics, epigenomics, transcriptomics, proteomics and metabolomics, the introduction of novel omics techniques may allow for the identification of novel biomarkers in ESRD. Several studies have suggested the potential role of bile acids in chronic kidney disease (*Marečková et al., 1990*; *Jimenez et al., 2002*; *Chu et al., 2015*). However, whether bile acid metabolism is involved in the metabolic disorders of ESRD and the impacts of bile acids on the prognosis of ESRD are still obscure. Thus, the purpose of this study was to investigate the relationship between altered bile acid profile and the prognosis of ESRD patients receiving hemodialysis. Ultra performance liquid chromatography-tandem mass spectrometry (UPLC-MS/MS), a well-adopted technique in metabolomics research, was used to quantitatively analyze the bile acid profile of ESRD patients on regular hemodialysis.

## SUBJECTS AND METHODS

### Patient recruitment and sample collection

A total of 77 ESRD patients (43 males and 34 females) who underwent regular hemodialysis at the Department of Nephrology, the Second Affiliated Hospital of Chongqing Medical University from December 2013 to December 2016 were recruited, and followed up for a long period.

Patients were eligible for this study if they were aged between 40 and 70 years, had no previous infections, received no antibiotic treatment within the past three months, had glomerular filtration rate <15 mL/min/1.73 m$^2$ (calculated by the Cockcroft-Gault formula), and underwent regular hemodialysis two to three times per week. Exclusion criteria were cholestasis or abnormal markers (serum or imaging) for liver function, history of viral hepatitis, alcoholic liver disease or steatohepatitis that may cause liver damage, history of gallbladder disease or gallbladder removal, and concomitant active malignant tumor.

Meanwhile, 30 age- and gender-matched healthy subjects (17 males and 13 females) were selected as a control group from the physical examination center of the Second Affiliated Hospital of Chongqing Medical University. Fasting blood samples were collected from all subjects, and transported on ice to the laboratory within 2 h of collection. Subsequently, the blood components were separated, aliquoted and stored at $-80\,^{\circ}$C until further analyses. Patient data such as age, gender, height and weight were collected, and the body mass index (BMI) was calculated as follows: ((weight (kg)/height$^2$ (m$^2$)). The present study was approved by the Ethics Committee of the Second Affiliated Hospital of Chongqing Medical University, the approval number was (2018)028. Written informed consent was obtained from all subjects prior to enrollment.

### Materials and reagents

High performance liquid chromatography-mass spectrometry (HPLC-MS) grade methanol was purchased from Fisher Scientific (Hampton, NH, USA). Both formic acid (HPLC grade) and ammonium acetate (HPLC-MS grade) were purchased from Sigma-Aldrich (St. Louis, MO, USA). The authentic standards of both Uncon and ConBA were purchased from Steraloids (Newport, RI, USA). Four deuterium-labeled bile acids (CA-d4, CDCA-d4, LCA-d4 and GCA-d4) were purchased from Steraloids (Newport, RI, USA) and other six deuterium-labeled bile acids (DCA-d4, GCDCA-d4, GDCA-d4, UDCA-d4, GLCA-d4 and GUDCA-d4) were purchased from Cambridge Isotope Laboratories (Tewksbury, MA, USA). The 10 deuterium-labeled bile acids were used as isotope-labeled internal standards (ISs) for quantitation and identification of compounds. Other materials were purchased from Anpel Laboratory Technologies (Shanghai, China).

### Preparation of standard curves

Aliquots of each BA standard solution were mixed to generate a stock solution at 1,000 ng/mL in methanol. The stock solution was serially diluted with methanol to 400, 200, 100, 40, 20, 10, 4 and 2 ng/mL. The stock solutions of the 10 ISs were prepared with methanol, and further diluted to a concentration of 100 ng/mL using methanol.

To prepare a standard curve, the above nine stock solutions were isometrically mixed with 100 ng/mL of IS solution to generate calibration points covering a concentration range of 1–500 ng/mL for all the analytes.

## Serum samples preparation

A 50-μL aliquot of serum sample was mixed with 200 μL methanol/acetonitrile solution (5:3,v/v). After centrifugation for 15 min at 14,000 g and 4 °C, 200 μL supernatant was dried under a stream of nitrogen. The residues were re-dissolved in 100 μL of 50% aqueous methanol containing 50 ng/mL ISs prior to UPLC-MS/MS analysis, with an injection volume of 7.5 μL.

## UPLC-MS/MS analysis

The samples were analyzed by UPLC-MS/MS using an Acquity UPLC system (Waters Corporation, Milford, MA, USA) coupled with a Triple Quad$^{TM}$ 5500 tandem mass spectrometer (AB Sciex, Framingham, MA, USA). All the samples were injected onto a Waters UPLC BEH C18 column (100 × 2.1 mm, 1.7 μm) at a flow rate of 0.35 mL/min. The mobile phase consisted of 10 mM ammonium acetate, 0.012% formic acid in water (A) and methanol (B). The chromatographic separation was conducted by a gradient elution program as follows: 0–0.5 min, 35%B; 1 min, 60%B; 4 min, 80%B; 5.3–6 min, 100%B; and 6.1–8 min, 35%B. The column temperature was set to 45 °C.

The analytes eluted from column were ionized in a negative ion mode electrospray ionization (ESI-) mode. The source temperature was set at 550 °C. The pressures of curtain gas, ion source gas 1, ion source gas 2 and collision gas were 30, 50, 50 and 8 psi, respectively. The voltage of ion spray (IS) was −4,500V, while −10V for both entrance potential and collision cell exit potential. Multiple reaction monitoring (MRM) was used to acquire data under optimal conditions of MRM transition (precursor > product), declustering potential and collision energy, as described in Table S1. The optimal dwell time was eight ms. Both samples and standard curves samples were analyzed simultaneously. Analyst software (version 1.5.2; AB Sciex, Framingham, MA, USA) was used for instrument control and data acquisition.

## Mass spectrometry data pretreatment

Mass spectrometry data were also analyzed by Analyst version 1.5.2 software (AB Sciex). The default parameters and assisting manual inspection were adopted to ensure the qualitative and quantitative accuracies of each compound. The peak areas of target compounds were integrated and output for quantitative calculation. Considering the extensive individual variations of BA concentrations, the relative proportion of total bile acid (TBA) was calculated for comparison. TBA was defined as a sum of all the 26 BAs associated with ESRD risk. The proportion of each BA was calculated as follow: concentrations of each BA/TBA * 100% (*Yu et al., 2015*).

## Statistical analysis

Statistical analyses were performed using SPSS 24.0 software (SPSS, Chicago, IL, USA). Continuous variables were tested for normal distribution via the Shapiro–Wilk test.

Mann–Whitney $U$-test or independent sample $t$-test was used to compare the metabolomic data between ESRD and HC groups. One-way analysis of variance followed by post hoc testing was used to compare the metabolomic data among ESRD death group, ESRD survival group and HC group. Least significant difference and Dunnett's T3 methods were applied to the absence and presence of heterogeneity of variance, respectively. $P$-value of less than 0.05 was considered statistically significant.

Multivariate analyses were performed with SIMCA-P+ 14.0 software (Umetrics AB, Umea, Sweden). Principal component analysis (PCA) was used for visual clustering and grouping, as well as initial evaluation of the model's validity. Supervised orthogonal partial least squares-discriminant analysis (OPLS-DA) was used to further compare the differences between the two groups, where the effects of model fitting were evaluated with R2X, R2Y and Q2Y model parameters. R2X and R2Y represent the fractions of the variance in X and Y, respectively, that are explained by OPLS-DA model, while Q2Y indicates the predictive performance of the model. Permutation test was used to verify the model in order to avoid over-fitting. Loading plots and variable importance of projection (VIP) > 1 were adopted as the criteria for screening differentially expressed bile acids. Next, bile acid components contributed most to the intergroup difference were identified using univariate statistical tests (*Kondo et al., 2014*; *Zhu et al., 2015*). Receiver operating characteristic (ROC) curve was plotted to calculate the area under the curve (AUC) in order to assess the diagnostic efficacies of the differential metabolites.

# RESULTS

## Demographic and clinical characteristics of patients

Demographic and clinical characteristics of 107 subjects are shown in Table 1. There were no significant differences between ESRD group and HC group in terms of age, BMI and sex ratio. With regard to laboratory indices, the levels of serum creatinine (CREA), blood urea nitrogen, uric acid, phosphorus, triglyceride (TG) and TBA were significantly elevated ($P < 0.05$) in ESRD patients compared to HC group, while the levels of blood calcium, albumin, total CHOL, high-density lipoprotein cholesterol (HDL-C), low-density lipoprotein cholesterol (LDL-C), hemoglobin and platelet count were significantly decreased ($P < 0.05$). These findings were consistent with the general clinical characteristics of ESRD patients. Nonetheless, there were no significant differences in the levels of alanine aminotransferase, aspartate aminotransferase (AST), alkaline phosphatase, total bilirubin and direct bilirubin between the two groups. Subgroup analyses revelaed that ESRD death group had increased mean age, reduced serum CREA level, and elevated levels of AST and TBA compared to ESRD survival group. However, no significant differences was found in other parameters between the two subgroups (Table 2).

## Univariate analyses of serum bile acid profile

The serum bile acid profiles of 77 ESRD patients and 30 HC were analyzed by UPLC-MS/MS. The differences in bile acid profile between ESRD group and HC group was compared by Mann–Whitney U test, and the results are shown in Table 3. A total of 32 bile acid

**Table 1 Demographic and clinical characteristics of ESRD and control group.**

| Variables | ESRD (N = 77) | HC (N = 30) | P-value |
|---|---|---|---|
| Age (year) | 59 ± 13.9 | 59.3 ± 6.8 | 0.956 |
| Sex (M/F) | 77 (43/34) | 30 (17/13) | 0.939 |
| BMI (kg/m$^2$) | 23 ± 3.2 | 22.1 ± 1.5 | 0.091 |
| PTH (pg/mL) | 246.7 ± 47.9 | – | – |
| CREA (μmol/L) | 905 ± 268.9[a] | 65.4 ± 14.7 | <0.001 |
| BUN (mmol/L) | 22.6 ± 6.2[a] | 5.2 ± 1.1 | <0.001 |
| UA (μmol/L) | 444.9 ± 110.8[a] | 284.3 ± 66.7 | <0.001 |
| Ca (mmol/L) | 2.2 ± 0.2[a] | 2.4 ± 0.2 | <0.001 |
| P (mmol/L) | 1.8 ± 0.5[a] | 1.0 ± 0.1 | <0.001 |
| TG (mmol/L) | 1.9 ± 1.1[a] | 1.1 ± 0.5 | <0.001 |
| CHOL (mmol/L) | 3.5 ± 1.0[a] | 4.7 ± 0.5 | <0.001 |
| HDL-C (mmol/L) | 1.0 ± 0.4[a] | 1.5 ± 0.3 | <0.001 |
| LDL-C (mmol/L) | 1.7 ± 0.7[a] | 2.8 ± 0.5 | <0.001 |
| WBC (10$^9$/L) | 5.7 ± 1.7 | 5.4 ± 1.3 | 0.416 |
| HGB (g/L) | 106 ± 17.2[a] | 140 ± 19 | <0.001 |
| PLT (10$^9$/L) | 151 ± 62.6[a] | 193.7 ± 50.6 | 0.003 |
| GLU (mmol/L) | 6.1 ± 2.8 | 5.4 ± 0.9 | 0.061 |
| ALB (g/L) | 38.3 ± 4.0[a] | 44.8 ± 3.0 | <0.001 |
| AST (U/L) | 20 ± 16.3 | 18.1 ± 7.5 | 0.548 |
| ALT (U/L) | 13.4 ± 8.3 | 15.7 ± 6.3 | 0.190 |
| ALP (U/L) | 96.2 ± 47.2 | 92.2 ± 28.8 | 0.675 |
| TBIL (μmol/L) | 6.9 ± 2.8 | 7.3 ± 2.3 | 0.459 |
| DBIL (μmol/L) | 3.3 ± 1.5 | 3.3 ± 1.9 | 0.920 |
| TBA (ng/mL) | 3,151.4 ± 270.0[a] | 1,849.2 ± 269.8 | 0.001 |

Notes:

Data represent means ± SD or means ± SEM. Comparisons performed via *t*-test (ESRD vs. HC).

[a] *P* < 0.05.

components in serum were quantified, of which 26 bile acids were significantly correlated with ESRD. The proportions of unconjugated bile acids (UnconBA), cholic acid (CA), chenodeoxycholic acid (CDCA), deoxycholic acid (DCA), HDCA, UDCA, α+ωMCA, γMCA, 7KLCA, 12KLCA and 6,7-diketoLCA were all decreased, while only the proportion of unconjugated βMCA was significantly increased (*P* < 0.05) in ESRD group compared to HC group. The proportion of LCA showed a downward trend, but the difference was not statistically significant (*P* = 0.062). Altogether, these results indicate that the proportions of most UnconBA are reduced in ESRD patients. In addition, the proportions of ConBA glycocholic acid (GCA), glycochenodeoxycholic acid (GCDCA), taurocholic acid (TCA), taurochenodeoxycholic acid (TCDCA), taurohyocholic acid (THCA), tauro α-muricholic acid (TαMCA) and tauroursodeoxycholic acid in ESRD group were significantly higher (all *P* < 0.05) than those in HC group. Nevertheless, the proportions of the remaining ConBA, glycodeoxycholic acid (GDCA), glycohyocholic acid, glycolithocholic acid, glycoursodeoxycholic acid, taurodeoxycholic acid and tauro β-muricholic acid were not significantly different between the two groups.

**Table 2 Demographic and clinical characteristics of ESRD survival, ESRD death and control group.**

| Variables | ESRD survival ($N = 30$) | ESRD death ($N = 17$) | HC ($N = 30$) |
|---|---|---|---|
| Age (year) | 55.9 ± 13.5[b] | 70.7 ± 9.0[a] | 59.3 ± 6.8 |
| Sex (M/F) | 30 (17/13) | 17 (9/8) | 30 (17/13) |
| Duration (months) | 30 ± 21 | 29 ± 22 | – |
| BMI (kg/m$^2$) | 23.8 ± 3.2 | 21.5 ± 3.1 | 22.1 ± 1.5 |
| PTH (pg/mL) | 338.0 ± 104.1 | 240.8 ± 82.7 | – |
| CREA (μmol/L) | 939.3 ± 301.5[a,b] | 731.1 ± 206.8[a] | 65.4 ± 14.7 |
| BUN (mmol/L) | 22.2 ± 5.4[a] | 21.4 ± 7.5[a] | 5.2 ± 1.1 |
| UA (μmol/L) | 444.0 ± 99.6[a] | 442.4 ± 113.6[a] | 284.3 ± 66.7 |
| Ca (mmol/L) | 2.2 ± 0.2[a] | 2.2 ± 0.3[a] | 2.4 ± 0.2 |
| P (mmol/L) | 1.7 ± 0.4[a] | 1.6 ± 0.6[a] | 1.0 ± 0.1 |
| TG (mmol/L) | 2.2 ± 1.4[a] | 2.1 ± 1.1[a] | 1.1 ± 0.5 |
| CHOL (mmol/L) | 3.9 ± 0.8[a] | 3.7 ± 1.0[a] | 4.7 ± 0.5 |
| HDL-C (mmol/L) | 1.0 ± 0.4[a] | 1.0 ± 0.4[a] | 1.5 ± 0.3 |
| LDL-C (mmol/L) | 2.1 ± 0.7[a] | 1.7 ± 0.8[a] | 2.8 ± 0.5 |
| WBC (10$^9$/L) | 5.6 ± 1.4 | 5.7 ± 1.8 | 5.4 ± 1.3 |
| HGB (g/L) | 112.5 ± 16.0[a] | 103.8 ± 20.8[a] | 140 ± 19 |
| PLT (10$^9$/L) | 171.6 ± 67.1 | 144.2 ± 72.4[a] | 193.7 ± 50.6 |
| GLU (mmol/L) | 6.1 ± 3.5 | 6.9 ± 2.6 | 5.4 ± 0.9 |
| ALB (g/L) | 38.9 ± 4.2[a] | 38.4 ± 4.3[a] | 44.8 ± 3.0 |
| AST (U/L) | 15.6 ± 6.9[b] | 22.4 ± 11.4 | 18.1 ± 7.5 |
| ALT (U/L) | 12.8 ± 7.1 | 14.9 ± 9.5 | 15.7 ± 6.3 |
| ALP (U/L) | 95.1 ± 49.0 | 113.1 ± 50.5 | 92.2 ± 28.8 |
| TBIL (μmol/L) | 6.4 ± 1.8 | 7.2 ± 2.3 | 7.3 ± 2.3 |
| DBIL (μmol/L) | 3.2 ± 1.0 | 3.7 ± 1.2 | 3.3 ± 1.9 |
| TBA (ng/mL) | 2,888.83 ± 373.4[a,b] | 4,213.0 ± 785.7[a] | 1,849.2 ± 269.8 |

Notes:
Data represent means ± SD or means ± SEM. Comparisons performed via one-way ANOVA.
[a] $P < 0.05$ vs. HC cohort.
[b] $P < 0.05$ vs. ESRD death cohort.

To further analyze the changing patterns of bile acids, all the bile acid components were classified according to their chemical properties and subsequently compared. The results showed that ESRD patients exhibited lower proportions of total UnconBA and secondary bile acids (SBA), while higher proportions of total ConBA, PBA, taurine-conjugated bile acids (TaurineBA) and glycine-conjugated bile acids compared to HC group ($P < 0.05$).

## Multivariate analyses of serum bile acid profile

The distribution of bile acids in the samples was preliminarily investigated by PCA in order to verify the rationality of experimental design and the homogeneity of biological replicates. The results of PCA are presented as a two-dimensional score plot (Fig. 1A), where each point represents a serum sample. Notably, a high differentiation was observed between ESRD group and HC group (R2X = 0.645, Q2 = 0.308), with a partial overlap and the top four principal components accounted for 64.5% of the total variance.

**Table 3  The percentages of individual bile acid relative to TBA in ESRD and healthy control group (%).**

| Metabolite | ESRD ($N = 77$) | HC ($N = 30$) | Z-value | P-value |
|---|---|---|---|---|
| CA | 1.32 (0.71–2.82) | 2.43 (1.06–6.56) | 2.927 | 0.003 |
| CDCA | 3.86 (1.88–8.10) | 14.76 (6.49–22.74) | 5.049 | <0.001 |
| DCA | 1.37 (0.21–4.37) | 6.50 (1.39–17.28) | 3.544 | <0.001 |
| HDCA | 0.13 (0.05–0.38) | 0.51 (0.08–1.72) | 2.592 | 0.010 |
| LCA | 0.10 (0.02–0.25) | 0.26 (0.00–0.64) | 1.865 | 0.062 |
| UDCA | 0.48 (0.17–1.06) | 1.17 (0.71–1.89) | 3.024 | 0.002 |
| α+ωMCA | 0.10 (0.03–0.29) | 0.39 (0.13–0.93) | 2.871 | 0.004 |
| βMCA | 0.05 (0.00–0.17) | 0.00 (0.00–0.00) | −3.700 | <0.001 |
| γMCA | 0.48 (0.15–1.05) | 0.93 (0.50–2.23) | 3.329 | 0.001 |
| 7KLCA | 0.22 (0.12–0.48) | 0.48 (0.28–0.70) | 3.440 | 0.001 |
| 12KLCA | 0.07 (0.00–0.22) | 0.36 (0.00–1.03) | 2.646 | 0.008 |
| 6,7-diketoLCA | 0.08 (0.00–0.22) | 0.23 (0.04–0.40) | 2.330 | 0.020 |
| GCA | 11.06 (8.22–14.88) | 6.13 (4.04–8.76) | −4.383 | <0.001 |
| GCDCA | 41.62 (33.90–48.88) | 30.44 (25.60–38.81) | −4.016 | <0.001 |
| GDCA | 2.59 (0.20–8.16) | 5.53 (0.97–11.10) | 1.298 | 0.194 |
| GHCA | 0.52 (0.32–0.90) | 0.48 (0.35–0.76) | −0.194 | 0.846 |
| GLCA | 0.02 (0.00–0.20) | 0.09 (0.00–0.26) | 0.813 | 0.416 |
| GUDCA | 8.79 (4.93–13.32) | 6.67 (2.76–11.00) | −1.345 | 0.178 |
| TCA | 2.52 (1.66–3.77) | 1.13 (0.61–1.57) | −5.229 | <0.001 |
| TCDCA | 5.33 (3.69–9.71) | 2.56 (1.79–3.88) | −4.834 | <0.001 |
| TDCA | 0.67 (0.09–1.68) | 0.74 (0.16–1.44) | 0.097 | 0.922 |
| THCA | 0.13 (0.08–0.26) | 0.11 (0.06–0.15) | −2.057 | 0.040 |
| TαMCA | 0.10 (0.06–0.16) | 0.05 (0.00–0.07) | −4.483 | <0.001 |
| TβMCA | 0.45 (0.27–0.71) | 0.42 (0.24–0.67) | −0.388 | 0.698 |
| TUDCA | 0.63 (0.36–1.41) | 0.33 (0.24–0.67) | −3.593 | <0.001 |
| UnconBA | 12.22 (6.77–20.86) | 35.65 (22.65–52.91) | 6.124 | <0.001 |
| TaurineBA | 11.38 (8.05–17.91) | 5.78 (4.21–8.42) | −4.459 | <0.001 |
| GlycineBA | 71.15 (65.22–80.59) | 54.54 (42.78–66.70) | −4.862 | <0.001 |
| ConBA | 87.78 (79.14–93.23) | 64.35 (47.09–77.35) | −6.124 | <0.001 |
| PBA | 74.95 (65.37–83.39) | 65.49 (54.34–74.97) | −3.274 | 0.001 |
| SBA | 20.43 (12.13–29.42) | 26.20 (17.70–36.79) | 2.303 | 0.021 |
| PBA/SBA | 3.63 (2.15–7.09) | 2.65 (1.44–4.40) | −2.538 | 0.011 |
| UnconBA/ConBA | 0.14 (0.07–0.26) | 0.55 (0.29–1.13) | 6.124 | <0.001 |

**Notes:**
Data presented as median and inter-quartile range (IQR).
Abbreviations: CA, cholic acid; CDCA, chenodeoxycholic acid; DCA, deoxycholic acid; HDCA, hyodeoxycholic acid; LCA, lithocholic acid; UDCA, ursodeoxycholic acid; α+ωMCA, α-muricholic acid and ω-muricholic acid; βMCA, β-muricholic acid; γMCA, γ-murocholic acid; 7KLCA, 7-ketolithocholic acid; 12KLCA, 12-ketolithocholic acid; 6,7-diketoLCA, 6,7-diketolithocholic acid; GCA, glycocholic acid; GCDCA, glycochenodeoxycholic acid; GDCA, glycodeoxycholic acid; GHCA, glycohyocholic acid; GLCA, glycolithocholic acid; GUDCA, glycoursodeoxycholic acid; TCDCA, taurochenodeoxycholic acid; TDCA, taurodeoxycholic acid; THCA, taurohyocholic acid; TαMCA, tauro α-muricholic acid; TβMCA, tauro β-muricholic acid; TUDCA, tauroursodeoxycholic acid; UnconBA, unconjugated bile acid; ConBA, conjugated bile acid; PBA, primary bile acid; SBA, secondary bile acid. TaurineBA, taurine conjugated bile acid; GlycineBA, glycine conjugated bile acid.

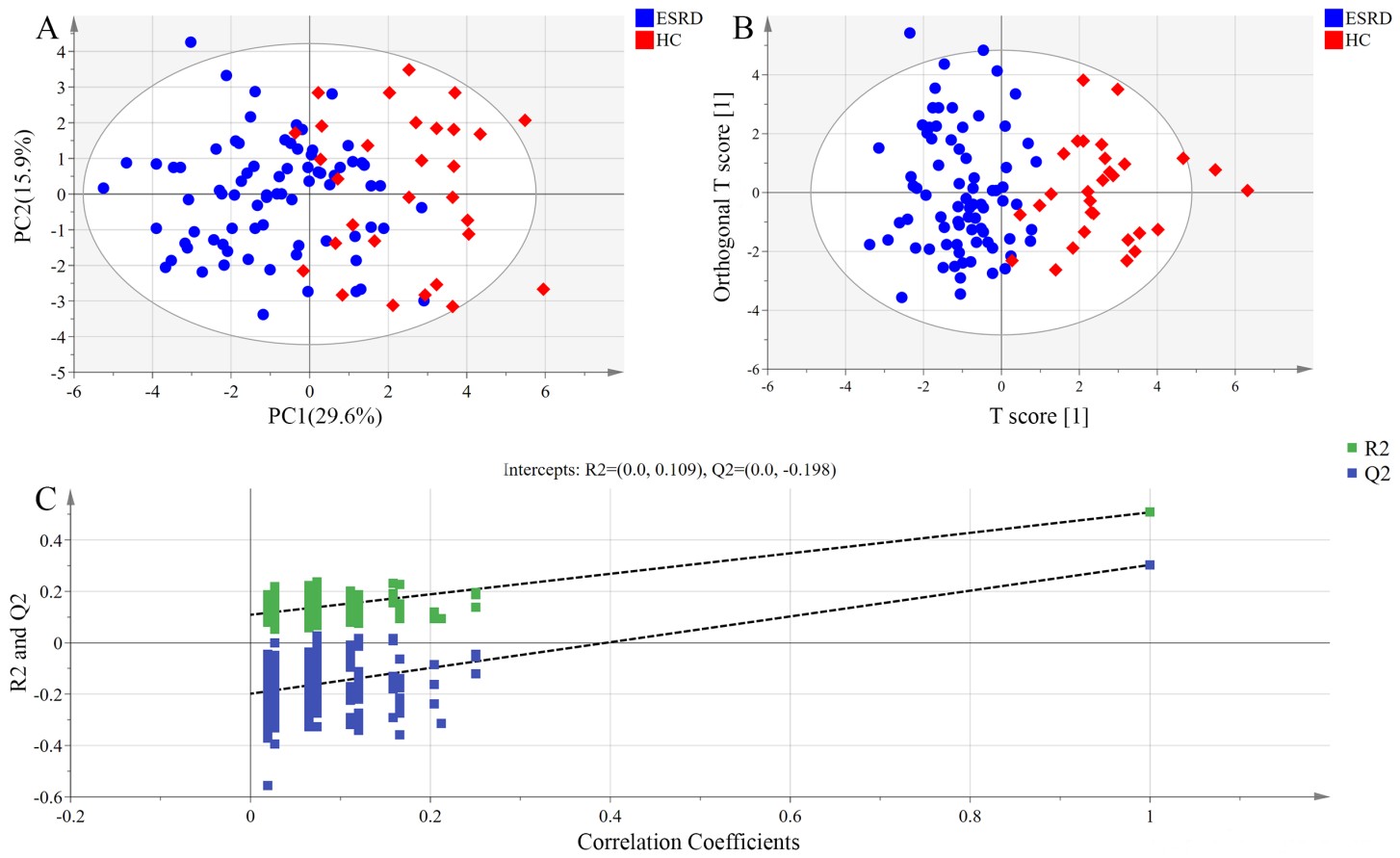

**Figure 1 Metabolomic analysis of serum samples.** (A) PCA score plots. Ellipse:Hotelling's T2 (95%). (B) OPLS-DA score plots showing a clear discrimination between ESRD (blue sphere) and HC subjects (red octahedron). Ellipse:Hotelling's T2 (95%). (C) Permutation test demonstrating the robustness of OPLS-DA model. R2 = 0.109 and Q2 = −0.198.

Orthogonal partial least squares-discriminant analysis model exhibited higher ability for intergroup comparisons than PCA model. Thus, OPLS-DA scores were plotted to further analyze the difference between ESRD group and HC group, and to identify the variables contributed most to the difference between the two groups. As shown in Fig. 1B, the serum bile acid profiles of the two groups displayed an obvious separation trend in OPLS-DA model, with the model parameter values of R2Xcum = 0.564, R2Ycum = 0.508 and Q2Ycum = 0.303. These data indicate that the bile acid metabolic profiles of the two groups may manifest as different phenotypes.

The results of the permutation test used for validating the OPLS-DA model are presented in Fig. 1C. R2 denotes the explanation capacity of the model, while Q2 represents the predictive capacity of the model. The low values of intercepts, R2 (0.109) and Q2 (−0.198), revealed that the constructed model was not over-fitting.

The most significant variables were visually displayed by a loading plot. The variables with statistical significance in univariate analysis ($P < 0.05$) and VIP > 1 were used as the criteria for the identification of differentially expressed bile acids. A total of six bile acids (GCDCA, CDCA, DCA, TCDCA, GCA and CA) were identified as potential biomarkers for differentiating ESRD patients from healthy subjects (Table 4).

**Table 4 Potential biomarkers discovered by VIP value and ROC analysis.**

| Biomarkers | VIP | P-value | AUC |
|---|---|---|---|
| GCDCA | 2.48 | <0.001 | 0.751 |
| CDCA | 2.20 | <0.001 | 0.815 |
| DCA | 2.06 | <0.001 | 0.721 |
| TCDCA | 1.41 | <0.001 | 0.802 |
| GCA | 1.38 | <0.001 | 0.774 |
| CA | 1.10 | 0.003 | 0.683 |

**Note:**

Variable important in projection (VIP) was obtained from OPLS-DA with a threshold of 1.0; $P$-value was calculated from Mann–Whitney $U$ test; Area under curve (AUC) was obtained from ROC curves.

### ROC curve analysis of differentially expressed bile acids

The six differentially expressed bile acids and TBA screened by OPLS-DA model were presented graphically as ROC curves in order to explore their diagnostic abilities. As shown in Fig. 2, the differentiating ability of TBA (AUC = 0.701, (95% CI [0.591–0.810]); $P = 0.001$) was lower than that of the six identified bile acids (except for CA). Among the six bile acids, CDCA (AUC = 0.815, (95% CI [0.726–0.904]); $P < 0.001$) exhibited the highest ability to differentiate ESRD patients from healthy subjects. In addition, a combination of all six differential bile acids demonstrated the most distinguishing ability (AUC = 0.890, (95% CI [0.818–0.963]); $P < 0.001$), hence can be used as a combined biomarker for ESRD diagnosis.

### Analysis of bile acid compositions among the subgroups

To further assess the differences in bile acids among ESRD patients with different prognoses, we retrospectively analyzed the survival rates of 47 patients whose samples were collected during 2013, and found that 17 patients died over a period of 3 years. Therefore, we established two subgroups consisted of 17 dead patients and 30 survivors, and then compared with HC group. The results showed that the abundances of TCA, TCDCA, THCA, TαMCA, TaurineBA and TBA in the control group, survival group and death group increased successively (Fig. 3), with a statistically significant difference among the two groups ($P < 0.1$). Altogether, these findings indicate that poor prognosis and mortality are positively correlated to the concentrations of the above-mentioned bile acids.

### DISCUSSION

In this study, we quantitatively analyzed the serum bile acid profiles in ESRD patients and HC using UPLC-MS/MS in order to determine the roles of bile acids in ESRD. The results showed that ESRD patients and HC displayed significant alterations in the proportions of bile acids. Notably, the levels of TBA were significantly increased in ESRD patients, as consistent with the findings of previous studies (*Balestri & Cupisti, 1996*). Previous studies have shown that the increased TBA level in ESRD patients is probably caused by subclinical liver damage, abnormal enterohepatic circulation (*Jimenez et al., 2002*), abnormal reabsorption and secretion of bile acids by renal tubule, and diminished excretion of bile acids from kidney due to low glomerular filtration rate (*Chu et al., 2015*).

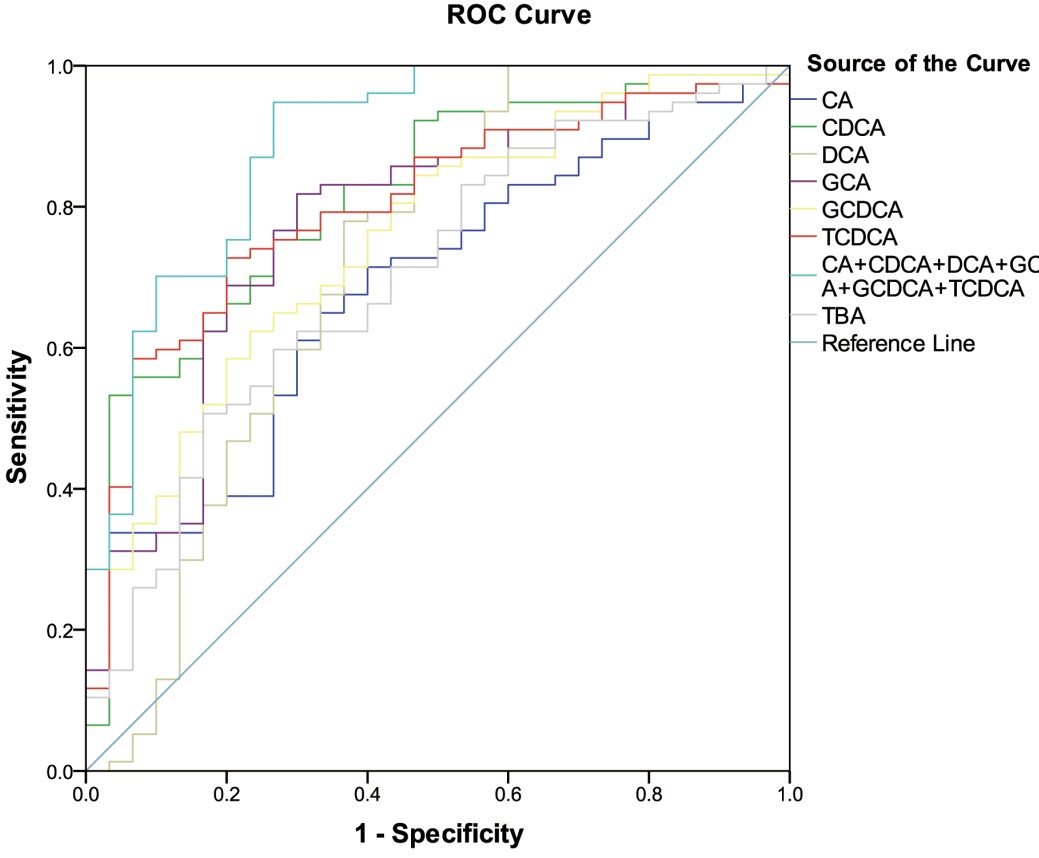

**Figure 2 ROC curves for bile acid biomarkers.** ROC curves analysis for the predictive power of six bile acid biomarkers and TBA for distinguishing ESRD from healthy control. The final logistic model included six bile acid biomarkers. Using a cutoff probability of 50%, we obtained sensitivity of 94.8% and specificity of 73.3% by ROC. The calculated area under the ROC curve was 0.890 (95% confidence intervals [0.818–0.963]).

In addition, *Gai et al. (2014)* suggest that the elevated plasma BA level represents an early event in the progression of chronic renal failure, and is due to increased efflux across the basolateral membrane of human hepatocytes.

The roles of bile acids in the body metabolism and disease onset are difficult to understand by the sheer investigative effort involved in the altering levels of TBA due to their complex bile acids metabolism and diverse chemical structures. Therefore, we classified each bile acid according to its chemical property for the purpose of comparison, and found that ESRD patients exhibited decreased proportions of UnconBA and SBA, as well as increased proportions of ConBA and PBA. Increasing evidence has shown that intestinal microbiota regulates the conversion of PBA to SBA, which in turn affects the size and composition of bile acid pool (*Duboc et al., 2013*). After the hepatic PBA enter the intestinal tract, the hydrolysis of the amino bond at the 24th carbon atom of bile acid conjugates is catalyzed by bile salt hydrolase (produced by intestinal microbiota) to form UnconBA. This reaction is followed by 7α dehydroxylation to form SBA. However, only the free bile acid salts can carry out 7α dehydroxylation. Hence, the conversion of ConBA to free bile acids by bile salt hydrolase is the premise of 7α dehydroxylation

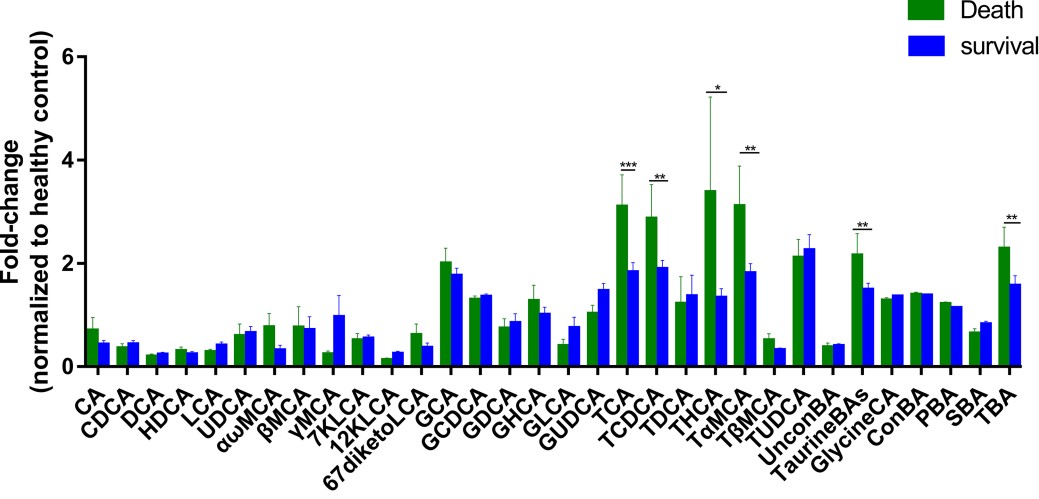

**Figure 3 Differential serum bile acid profiling among the two cohorts.** The fold-changes in bile acid levels between ESRD death and survival groups (relative to the control cohort) are graphically displayed. Normally-distributed data are depicted in a bar graph with means and standard errors of the mean (SEMs). $^*P < 0.1$; $^{**}P < 0.05$; $^{***}P < 0.01$.

(*Begley, Gahan & Hill, 2005*). Our results suggest that the decreased proportions of UnconBA and increased proportions of ConBA in ESRD patients may be related to the low activity of bile salt hydrolase resulted from the disordered intestinal microbiota and diminished hydrolysis of bile acids in the intestinal tract. This may lead to an increase in the synthesis of PBA and a decrease in the synthesis of SBA.

To further analyze the bile acid compositions that play key roles in the onset and progression of ESRD, six typical differential bile acids that differentiate ESRD patients from HCs were screened and identified by OPLS-DA model, including CDCA, DCA, CA, GCDCA, TCDCA and GCA. ROC analysis revealed that the combination of the above six bile acids was superior to total or individual bile acids for detecting metabolic complications in ESRD patients. Notably, these six bile acids are mainly responsible for the abnormal changes in bile acid metabolism in ESRD patients.

Bile acids have been well known to act as emulsifiers to promote the absorption of lipids and fat-soluble vitamins in the small intestine, but the knowledge of bile acids has been constantly evolving over the last 20 years. They were recently found to act as endogenous signaling molecules in mammals for activating FXR, GPBAR and cell signaling pathways in vivo. Activation of these receptors and cell signaling pathways is involved in the regulation of many important life processes, including the synthesis, metabolism and transport of bile acids (*Ananthanarayanan et al., 2001*; *Goodwin et al., 2000*; *Pircher et al., 2003*), blood glucose (*Katsuma, Hirasawa & Tsujimoto, 2005*; *Stayrook et al., 2005*), and lipoprotein (*Hirokane et al., 2004*; *Watanabe et al., 2004*), as well as energy metabolism (*Watanabe et al., 2006*).

Farnesoid X receptor is the most important bile acid receptor, not only expressed in the enterohepatic circulation, but also in the heart and kidneys (*Forman et al., 1995*; *Kawamata et al., 2003*). UnconBA such as CDCA, DCA, LCA and CA are a group of

FXR agonists with high affinity, among which CDCA possesses the highest (*Jia, Xie & Jia, 2017*; *Makishima et al., 1999*; *Song et al., 2015*; *Vaquero et al., 2013*). FXR activation was shown to reduce circulating TG levels by lowering the production of very LDL and increasing its clearance (*Watanabe et al., 2004*). Moreover, the serum levels of TG and HDL-C were increased in FXR-deficient mice (*Lefebvre et al., 2009*). The enterohepatic circulation of bile acids is disturbed in hyperlipidemia patients receiving bile acid sequestrants or patients undergoing ileectomy, thus leading to an increase in plasma TG level (*Angelin et al., 1978*; *Buchwald et al., 1990*). Interestingly, oral administration of CDCA contributes to a significantly greater reduction in TGs compared to placebo (*Carulli et al., 2013*). Likewise, our study found that the levels of CDCA, DCA and CA were decreased in ESRD patients, suggesting that the inhibition of FXR activity may increase plasma TG levels, which represents one of the mechanisms of dyslipidemia in ESRD patients.

In addition, our study found that the serum levels of CHOL and LDL-C decreased remarkably in ESRD patients (Table 1). The serum CHOL levels in patients with chronic renal failure and patients undergoing hemodialysis may be different from those in healthy population. The lower the serum CHOL level (50–150 mg/dl), the higher the mortality rate, so-called a "reverse epidemiology" phenomenon (*Kalantar-Zadeh et al., 2003*). The activation of FXR and its related signaling pathways has been found to reduce bile acid synthesis by decreasing the biological activity of rate-limiting enzyme CYP7A1, and leading to an increase in serum CHOL level (*Lefebvre et al., 2009*). It is speculated that the FXR activity in ESRD patients can be inhibited due to the decreased expression of FXR agonists, which may explain the lower levels of serum CHOL and LDL-C in ESRD patients compared to HC.

Subgroup analysis was performed on ESRD patients with differing prognoses and HC, and the results showed that the abundances of TCA, TCDCA, THCA, TαMCA, TaurineBA and TBA were significantly increased in ESRD death group compared to ESRD survival group (Fig. 3). Noticeably, the above bile acids with high abundances were all TaurineBA. These findings demonstrate that the increased TaurineBA levels are positively correlated to the poor prognosis of ESRD patients. Besides, the increased levels of TaurineBA could also promote the absorption of lipids in the small intestine, and increase the risk of hyperlipidemia and subsequent cardiovascular diseases. It has been reported that free taurine maintains the barrier function of intestinal epithelia by promoting NLRP6 inflammasome activation and enhancing intestinal epithelial IL-18 production (*Levy et al., 2015*). Therefore, it is speculated that the elevated levels of TaurineBA can reduce the concentrations of free taurine in ESRD patients, leading to an impaired intestinal epithelia barrier function and increased enterogenous urotoxin accumulation.

Of the six bile acids associated with ESRD risk, only TCDCA was differentially expressed between ESRD death group and ESRD survival group. The other metabolites (i.e., CA, CDCA, DCA, GCA and GCDCA) were not significantly different between ESRD death and survival groups. These results indicate that these metabolic biomarkers have no role in the prognosis of ESRD. It is postulated that abnormal bile acid metabolism occurs during the early stage of chronic kidney disease, which does not aggravate ESRD

progression and prognosis. The significant roles of TCDCA in the diagnosis and prognosis of ESRD patients remain largely unknown, which warrants further studies.

There are still some shortcomings in this study. First, the study subjects were selected from a single center, thus the sample size was relatively small. Second, in the subgroup analyses, the average age of ESRD patients who died during the 3-year follow-up was older than that of ESRD survival group, which may influence the findings of bile acid compositions between the two subgroups. Third, the mechanism of bile acid profile change affecting the prognosis of ESRD patients was speculated on the basis of the existing researches. Fourth, the elevated levels of ConBA in ESRD patients may be a consequence of imbalanced gut microbiota, altered enterohepatic circulation and/or hepatic dysfunction. Thus, further in vitro, in vivo and well-designed clinical studies are warranted to confirm the usefulness of bile acids in ESRD diagnosis and prognosis. Despite these limitations, this is the first study that utilizes metabolomics approach to identify a panel of bile acid biomarkers for the diagnosis and prognosis of ERSD. Given that the currently available diagnostic methods can only examine the symptoms of ESRD, this metabolomics approach is more appropriately used to describe the etiological mechanisms of the disease.

## CONCLUSION

In conclusion, this study reveals that the serum bile acid profile of ESRD patients is significantly different from that of healthy subjects. In addition, the altered serum bile acid profile may affect the occurrence of metabolic complications and prognosis of ESRD patients. Therefore, targeting bile acid compositions may provide a novel therapeutic strategy for the treatment of relevant metabolic complications in ESRD patients.

### Funding

This work was supported by the China Postdoctoral Science Foundation funded project (No. 2018M633630XB) and the Science and Technology Research Program of Chongqing Municipal Education Commission (Grant No. KJQN201800403). The funders had no role in study design, data collection and analysis, decision to publish, or preparation of the manuscript.

### Grant Disclosures

The following grant information was disclosed by the authors:
China Postdoctoral Science Foundation funded project: 2018M633630XB.
Science and Technology Research Program of Chongqing Municipal Education Commission: KJQN201800403.

### Competing Interests

The authors declare that they have no competing interests.

## Author Contributions

- Rong Li conceived and designed the experiments, performed the experiments, analyzed the data, contributed reagents/materials/analysis tools, prepared figures and/or tables, authored or reviewed drafts of the paper.
- Li Zeng conceived and designed the experiments, performed the experiments, contributed reagents/materials/analysis tools, authored or reviewed drafts of the paper.
- Shuqin Xie analyzed the data, contributed reagents/materials/analysis tools.
- Jianwei Chen prepared figures and/or tables.
- Yuan Yu prepared figures and/or tables, approved the final draft.
- Ling Zhong conceived and designed the experiments, authored or reviewed drafts of the paper, approved the final draft.

## Human Ethics

The following information was supplied relating to ethical approvals (i.e., approving body and any reference numbers):

The present study was approved by the Ethics Committee of the Second Affiliated Hospital of Chongqing Medical University (Chongqing, China) (approval number: (2018)028).

## Data Availability

The raw data are available in the Supplemental Files.

## Supplemental Information

Supplemental information for this article can be found online at http://dx.doi.org/10.7717/peerj.7145#supplemental-information.

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
