# Peer review of "Targeted metabolomics study of serum bile acid profile in patients with end-stage renal disease undergoing hemodialysis"

_PeerJ, doi:10.7717/peerj.7145_

## Round 0.1 · original submission · Major Revisions

Dear Dr. Zhong,

Your manuscript entitled "Targeted metabolomics study of serum bile acid profile in patients with end-stage renal disease undergoing hemodialysis" which you submitted to PeerJ, has been reviewed by the editor and 2 experts in the field.

I regret to inform you that the reviewers have raised serious concerns, and therefore your paper cannot be accepted for publication in PeerJ in its present form. However, since reviewers felt the manuscript contained some potentially interesting data, I would be willing to reconsider if you wish to undertake major revisions and resubmit.

If you decide to resubmit the revised version, please summarize all the improvements made in the new version and give answers to all critical points raised in the reviewers’ report in an accompanying letter. Please copy and paste each and every reviewer's comment above your response. If you feel any of their points are inappropriate, you are certainly free to provide rebuttal in your covering letter.

I strongly suggest providing a more detailed description of the method section, addressing the statistical issues raised by Reviewer #1, correcting mistakes on data presentation and improving language to ensure that your international audience can clearly understand your text. Moreover, the discussion should be more open to alternative explanations other than altered microbiota. Finally, the Authors may express opinions but all conclusions must be supported by the results. Speculation is welcomed but should be restricted to the discussion and clearly identified as such.

Please note that resubmitting your manuscript does not guarantee eventual acceptance. Since the requested changes are major, the revised manuscript will undergo a second round of review by the same reviewers. I must emphasize that the acceptability of the revision will depend upon the resolution of the points raised by the reviewers.

Sincerely yours,

Stefano Menini

·

Basic reporting

Several errors in punctuation and spelling need to be addressed. Specifically:
—Line 50: use a semi-colon or period before ‘…on the other hand…’
—Line 59: change it to ‘…are secreted’
—Line 91: change ‘[(’ to ‘[’; change ‘))’ at the end to ‘ )]’
—Line 96: change ‘Reagent’ to ‘Reagents’
—Line 97: change ‘...was the product of…’ to ‘…was purchased from…’
—Line 106: change ‘…curves samples’ to ‘…curves’
—Line 140: uncapitalize ‘…Extensive…’
—Line 166: use the correct plural form of ‘index’
—Line 166: ‘…baseline data and laboratory indexes…’ This is not an accurate description of the contents in Table 1. Just incorporate ‘Demographic and clinical characteristics’ into this sentence.
—Line 180: use capital T in ‘(table 2)’ as you previously did in Table 1. Add space in front of the parentheses.
—Line 184: what does ‘positively detected’ even mean? Change to a phrase that makes statistical sense.
—Line 215: ‘…had different phenotypes.’ This conclusion seem to be derived from the model parameters; however, these values cannot be considered as conclusive evidence that there is statistically significant differences between two phenotypes. Consider rephrasing, or provide evidence that these values indicate actual phenotype-level differences.
—Line 224: use capital T in ‘(table2)’.
—Line 262: change ‘amido’ to ‘amino’
—Line 280-283: This sentence is too long to comprehend in one read. For clarity sake, consider rephrasing or separating into multiple sentences.
—Line 310: ‘… showing an U-curve relationship)’ – what is the definition of a U-curve relationship? Where in your data is this shown?
—Line 312: ‘Bile acid is the main approach for cholesterol metabolism’. This sentence does not make logical sense with the previous sentence. Did you mean to start a separate paragraph here? Also, I do not understand what this sentence even means.

There are other problems with sentence constructions throughout the manuscript, makes it uneasy to comprehend. Overall English language revision of the total manuscript is desired. Authors should be more careful while preparing the manuscript. Besides typographical and English language mistakes, there are several mistakes on data presentation, as described below.

In this study, the authors addresses the relationship between changes in bile acid profiles and prognosis of End Stage Renal Disease (ESRD) in patients undergoing hemodialysis. Although the authors have briefly discussed importance of gut microbial metabolites in kidney disorders, they have not sufficiently justified why they are specifically interested in bile acids in their targeted metabolomics study.

Line 70 - ‘Previous studies have seldom focused on the expression of bile acid metabolic profile in chronic kidney disease.’ is not at all appropriate. There have been studies describing the role of bile acids in kidney diseases:
1. Marečková, O., Skala, I., Mareček, Z., Malý, J., Kočandrle, V., Schück, O. & Prat, V. (1990). Bile composition in patients with chronic renal insufficiency. Nephrology Dialysis Transplantation, 5(6), 423-425.
2. Jimenez, F., Monte, M. J., El-Mir, M. Y., Pascual, M. J., & Marin, J. J. G. (2002). Chronic renal failure-induced changes in serum and urine bile acid profiles. Digestive diseases and sciences, 47(11), 2398-2406.
3. Chu, L., Zhang, K., Zhang, Y., Jin, X., & Jiang, H. (2015). Mechanism underlying an elevated serum bile acid level in chronic renal failure patients. International urology and nephrology, 47(2), 345-351.

In the discussion section, authors do mention some of these studies, but state that the results of these studies are controversial (line 253). Is ‘controversial’ the correct term here? — Controversial according to whom?

Tables:
—What is superscript ‘a’ in Table 1?
—Abbreviations are missing for CA, CDCA, DCA, TCA,THCA,TalphaMCA, etc. in Table 3.
—p-values are shown as 0.000 in Table 4. Technically, this is incorrect based on how I presume the analysis was done; rather, put ‘<10^-3’.
—In the footnote of Table 4, correct the spelling error for ‘aculated’.

Figures:
—Correct the title of Figure 1. ‘Metabonomic’ should be ‘Metabolomic’ to be consistent with the contents of the manuscript.
—In Figure 1A, B and C, correctly label the names of the axes.
—What is R2, Q2 in figure 1, and also described in line 217? For those unfamiliar with OPLS-DA modeling, provide a brief explanation.
—I have no idea how to interpret Fig. 1C. Do the red and blue dots at X=1 indicate outliers? Make this more self-explanatory, as it’s hard to understand the main message of this figure.
—There are no legends in Figure 2. Please describe.
—It’s hard to understand what message authors want to convey by looking at Figure
3. The three groups are not clearly distinguishable. What are the differentially abundant bile acids?
—In the y-axis of Figure 3, change ‘Fold of Change’ to ‘Fold-change’; correct the spelling mistake of ‘heathey’.
—Make the bars in Figure 3 into color.
—If the fold-changes are relative to the control cohorts (as described in the first sentence of the legend), then the bar for ‘control’ is not necessary, as its height should always be 1.

Experimental design

– Methodology section does not provide sufficient information. For example, authors defined total bile acid (TBA) as a sum of 26 BAs (line 141). But it is not clear what are those BAs that the authors target.
– Why was multiple hypothesis correction not performed when comparing the groups?
– Authors should provide sufficient justification for using different types of statistical tests.
– Please explain parameters R2X, R2Y and Q2Y in OPLS-DA.
– Authors should clearly explain the subgroup analysis in a separate head in methodology section.

Validity of the findings

– In this manuscript, through a targeted metabolomics approach, the authors performed a comparative analysis of bile acid profiles of ESRD patients and healthy samples. Authors have identified six bile acids as biomarkers that can distinguish ESRD patients from healthy samples. However, as the authors did not perform any analysis on the gut microbiome composition of ESRD and healthy samples, repeatedly associating the observed differences in bile acid profiles to gut microbiome seems far-fetched. Consider truncating the speculative discussion on gut microbes.
– Line 315: ‘On the contrary, our study found that the FXR activity of ESRD patients was inhibited due to the decrease of FXR agonists…’ – where in your manuscript results have you checked FXR activity?
– It’s hard to find any clinical significance in distinguishing ESRD patients from healthy samples based on six bile acids biomarkers as proposed by authors. Also, the difference in bile acid profile may be due to the consequences of ESRD condition, which is needed to be verified. A more intriguing observation would have been to study whether these bile acids biomarkers can detect early stages of renal failure.
– Why the ROC analysis and subgroup analysis yield different sets of bile acids? The authors claim six bile acids as potential biomarkers to differentiate healthy and ESRD patients, but when they compare in three groups, they found different bile acids. Does that mean these biomarkers have no role in disease prognosis? Please clearly explain, as this pertains to one of the main messages of the manuscript.

Reviewer 2 ·

Basic reporting

Li et al., present a manuscript describing the serum bile acid profile in patients with end-stage renal disease. While these patients are known to have elevated total bile acids, this is the first report describing the specific bile acid species that are altered in disease. Moreover, the authors propose that these alterations may be due to the known disturbances in the gut microbiota. The manuscript is generally well written with the exception of some typographical errors and confusing phrases throughout.

Typographical errors throughout: For example:
line 28 “the” should be “The”,
line 59 “secret” should be “secrete”,
line 94, “were” should be “was”.

Line 68 “G protein-coupled receptor” should be “G protein-coupled bile acid receptor (GPBAR)” and GPBAR subsequently.
Line 195: it is unclear what “change rules” means.
Line 208: “variables” should be “variance”

Experimental design

The introduction and background are sufficient to orient the reader to the question being asked, and clearly states the purpose of the work. However, further introduction of current diagnostic methods for ESRD, and how this work may fill a diagnostic need and gap in knowledge should be addressed.

The authors present subgroup analysis data prior to telling the reader how these subgroups were obtained. With respect to this, the authors should not only present age differential between “death group” and “survival group”, but should provide further information regarding disease duration in patients for each group.

Validity of the findings

Several conclusions are not supported by the data. For example, the authors state that FXR activity is inhibited due to decreased concentration of agonists (line 315-318). However, FXR signaling was never assessed, for example by serum fgf19. The authors could also use 7aC4 test to assess Cyp7a activity more directly. Moreover, the authors should more clearly define other possible mechanisms for elevated serum bile acids including alterations in hepatic function, which is alluded to in the discussion. The following reference assesses bile acid transporters in rodent models of disease and may of use: Gai Z. 2014. Am J. Physiol. Renal Physiol.

The authors state as a conclusion that the increased level of taurine-conjugated bile acids… causing impaired barrier function and increased accumulation of urotoxin in ESRD patients (lines 328-331). The data do not support this conclusion and this should be stated as a speculation.

The authors should better explain how a bile acid profile improves upon current diagnostic methods (e.g., GFR).

The work presented provides rationale for hypothesis driven research, for example, the elevation of conjugated bile acids in ESRD patients may be the result of altered microbiota, but could also result from altered enterohepatic circulation and/or hepatic dysfunction, both testable hypothesis, this should be discussed with greater clarity.

Additional comments

The utility of the findings should be better explained. Does this work have the potential to change diagnostics in ESRD?
Speculations should be stated as such, not as conclusions from the work presented.
More detailed explanations as to why it is believed that these findings are not directly related to hepatic dysfunction as a consequence of ESLD.
Fix typographical errors/spelling mistakes throughout.

---

## Round 0.2 · Minor Revisions

Dear Dr. Zhong,

Your manuscript entitled "Targeted metabolomics study of serum bile acid profile in patients with end-stage renal disease undergoing hemodialysis" which you submitted to PeerJ, has been re-reviewed. The reviewer comments are included at the bottom of this letter and in the attached document.

I would urge you to give these points your careful attention.

I hope that you will be prepared to make the necessary amendments and submit a revised manuscript accompanied by a statement of how you have responded to the criticisms raised. Please copy and paste each and every reviewer's comment above your response.You are also kindly requested to provide a complete tracked changes version of the manuscript in order to make it easier to verify that the required changes have been made

I look forward to receiving your revision.

Sincerely yours,

Stefano Menini

·

Basic reporting

-In order to provide a more credible and professional contact email, can Professor Yuan Yu provide his/her professional email address? (i.e. ...@hospital.cqmu.edu.cn).
-Rather than having "P<" and "p<", please pick one and be consistent in the reporting.
-Line 297: Change "They are recently found" to "They were recently found".
-See title of Figure 1. My request to change "Metabonomic" to "Metabolomic" has not been addressed.
-Please remove the sentence "Bile acid synthesis is primarily involved in hepatic cholesterol metabolism, which causes an enhanced demand of cholesterol in the liver", as I don't believe that hepatic cholesterol metabolism is not the sole reason for an enhanced demand of cholesterol in the liver.
-See Table 1. Please keep the presentation of the superscripts consistent. You have "kg/m2", "10^9", and superscript a. I'd suggest changing everything to a true superscript.
-Please relabel the axes of the PCA plots in Fig. 1A. Simply replace t[1] and t[2] as PC1 and PC2, respectively (this is standard reporting for PCA plots). Same for Fig. 1B, as the general audience will not know what to[1] and t[1] will mean without any explanations in the legends. Please correctly label the x and y-axes. "200 permutations 1 components" is NOT the correct description of the measurement in the x-axis. Why are you just providing a screenshot? (I see the "SIMCA 14.1 - 2019/4/23 21:57:43...").
-See legend for Figure 2. "The curve is closet to the top left"...this does not make any sense. Please revise. There is enough standard text in the literature to use as a model for providing a basic description of an ROC curve.

Experimental design

no comment

Validity of the findings

no comment

Reviewer 2 ·

Basic reporting

The authors have extensively modified the text and greatly improved the language. However, a few minor typographical errors remain which can be easily addressed (e.g., metabolism is misspelled in line 59).

Experimental design

My concerns have been addressed.

Validity of the findings

My concerns have been adequately addressed.

---

## Round 0.3 · accepted · Accept

Dear Drs. Li and Zhong,

I am pleased to inform you that the revision of your manuscript entitled "Targeted metabolomics study of serum bile acid profile in patients with end-stage renal disease undergoing hemodialysis" now makes it acceptable for publication in PeerJ. I appreciate very much your making the suggested revisions. The manuscript will now be forwarded to the product editor for copy editing and publication.

I thank the reviewers for their effort in improving the manuscript and the authors for their cooperation throughout the review process.

Sincerely yours,

Stefano Menini

# ·

Basic reporting

No comment.

Experimental design

No comment.

Validity of the findings

No comment.

Additional comments

Thank you for taking the time to address all my concerns.